# PUSH: CONCURRENT PROBABILISTIC PROGRAMMING FOR BAYESIAN DEEP LEARNING

## ABSTRACT

We introduce a library called Push that takes a probabilistic programming approach to Bayesian deep learning (BDL). This library enables concurrent execution of BDL inference algorithms on multi-GPU hardware for neural network (NN) models. To accomplish this, Push introduces an abstraction that represents an input NN as a particle. Push enables easy creation of particles so that an input NN can be replicated and particles can communicate asynchronously so that a variety of parameter updates can be expressed, including common BDL algorithms. Our hope is that Push lowers the barrier to experimenting with BDL by streamlining the scaling of particles across GPUs. We evaluate the scaling behavior of particles on single-node multi-GPU devices on vision and scientific machine learning (SciML) tasks.

## 1 INTRODUCTION

Bayesian deep learning (BDL) (Blundell et al., 2015; Gal & Ghahramani, 2016; Kendall & Gal, 2017; Khan et al., 2018; Maddox et al., 2019; Wilson & Izmailov, 2020; Izmailov et al., 2021; Jospin et al., 2022) brings the benefits of Bayesian inference to deep learning. However, Bayesian inference over neural networks (NNs) can be computationally intensive and difficult to scale. We would thus like to lower the barrier to experimenting with novel BDL algorithms to better explore BDL's potential.

Probabilistic programming (Goodman et al., 2012; Mansinghka et al., 2014; Wood et al., 2014; Tran et al., 2017; Le et al., 2017; Huang et al., 2017; van de Meent et al., 2018; Tran et al., 2018; Bingham et al., 2019; Baudart et al., 2020; 2021; Lundén et al., 2022; Lew et al., 2023) is an approach to Bayesian inference where programming language and systems technology is applied to aid Bayesian inference. However, most existing probabilistic programming approaches do not focus on BDL. To address this gap, we introduce a library called Push that takes a probabilistic programming approach to BDL and describe its implementation (§ 4).

This library enables concurrent execution of BDL inference algorithms on multi-GPU hardware for neural network (NN) models. To accomplish this, Push introduces an abstraction that represents an input NN as a particle. Push enables easy creation of particles so that an input NN can be replicated and particles can communicate asynchronously so that a variety of parameter updates can be expressed, including common BDL algorithms such as SWAG (Maddox et al., 2019) and general-purpose inference algorithms such as Stein Variational Gradient Descent (SVGD) (Liu & Wang, 2016). Our hope is that Push lowers the barrier to experimenting with BDL by streamlining the scaling of particles across GPUs. We evaluate the scaling behavior of particles on single-node multi-GPU devices on vision and scientific machine learning (SciML) (Raissi et al., 2019; Karniadakis et al., 2021) tasks (§ 5).

## 2 RELATED WORK

Our work is inspired by probabilistic programming and deep probabilistic programming, although there are three main differences. First, most existing probabilistic programming approaches are ill-adapted for the use case of specifying priors on NNs. In particular, many approaches specify distributions on program traces (Wingate et al., 2011a;b; Goodman et al., 2012; Mansinghka et al.,

2014; Wood et al., 2014; Le et al., 2017; van de Meent et al., 2018) or use a restricted modeling language to provide notation to define familiar probabilistic models such as graphical models (Salvatier et al., 2016; Carpenter et al., 2017). The former approach is general which increases the difficulty of inference. The latter approach is less flexible. There are probabilistic programming approaches that focus on Gaussian processes (Schaechtle et al., 2016; Riutort-Mayol et al., 2023) which are equivalent to NNs under certain limits (Lee et al., 2017; Jacot et al., 2018). Push users directly work with vanilla NN models (§ 3.3).

Second, most existing probabilistic programming approaches do not focus on enabling BDL. Deep probabilistic programming (Tran et al., 2017; Shi et al., 2017; Tran et al., 2018; Bingham et al., 2019; Baudart et al., 2021) is a emerging paradigm that enables lightweight interoperability between probabilistic programs and NNs. However, these approaches focus on enabling Bayesian inference for typical probabilistic models and not Bayesian inference over NN priors. Our approach will focus on enabling BDL algorithms to address the gap in tool support for BDL (§ 3.4).

Third, most existing probabilistic programming approaches provide a separation of concerns between model and inference so that models are at a higher-level of abstraction compared to inference. The benefit of this approach is that the language/library implementation can fully automate and optimize inference. The drawback of this approach is that we lose flexibility in specifying inference algorithms which may be necessary for performing custom Bayesian inference on difficult probabilistic models. Our approach based on *particles* (§ 3.2) puts model and inference at the same level of abstraction, similar to some approaches to deep probabilistic programming (Tran et al., 2018). Thus, the focus of the library is on providing an efficient implementation of particles as opposed to full automation of inference.

## 3 PUSH LIBRARY

We introduce the Push library in this section. We begin by motivating the particle abstraction that Push provides (§ 3.1). We then introduce particles in Push (§ 3.2). Finally, we discuss how to define models in Push (§ 3.3) and demonstrate how it can be used to encode BDL algorithms (§ 3.4).

### 3.1 MOTIVATION FOR PARTICLES

The goal of Bayesian inference is to compute the posterior distribution $p(\theta|\mathcal{D}) = p(\mathcal{D}|\theta)p(\theta)/Z$ where $p(\mathcal{D}|\theta)$ is the likelihood of data $\mathcal{D}$ given parameters $\theta$, $p(\theta)$ is a prior distribution, and $Z = p(\mathcal{D})$ is a normalizing constant. In general, $Z$ is intractable to compute so that the posterior distribution is approximated. Variational inference (Hoffman et al., 2013; Ranganath et al., 2014; Blei et al., 2017) and Markov-Chain Monte Carlo (MCMC) (Gilks et al., 1995; Brooks et al., 2011) are two popular methods for approximating a posterior distribution. These techniques have been used to implement probabilistic programming languages (*e.g.*, see (Kucukelbir et al., 2015; Bingham et al., 2019; Lai et al., 2023) for variational inference and see (Goodman et al., 2012; Tristan et al., 2014; Carpenter et al., 2017) for MCMC).

Methods based on the concept of particles such as particle filtering (Djuric et al., 2003), Sequential Monte Carlo (SMC) (Doucet et al., 2001a;b), and SVGD (Liu & Wang, 2016) can also be used to approximate a posterior distribution. Particle-based methods have also been used to implement probabilistic programming languages (Wood et al., 2014; Lundén et al., 2022; Lew et al., 2023). A *particle* $\theta_i$ refers to a parameter setting of a distribution $p(\theta_i)$ and a set of particles $\{\theta_1, \ldots, \theta_n\}$ can be used to form a discrete approximation of the posterior $p(\theta|\mathcal{D})$ as $p(\theta|\mathcal{D}) \approx \frac{1}{n} \sum_{i=1}^{n} \delta_{\theta_i}(\theta)$ where $\delta_x$ is a Dirac delta at $x$. During inference, each particle is updated using $\mathcal{D}$ where the way in which a particle is used depends on the method (D'Angelo & Fortuin, 2021; Yashima et al., 2022). There are connections between particle-based methods and gradient flow (Liu, 2017; Fan et al., 2021).

In the setting of BDL, we use the terminology *particle* to refer to a NN. Thus, a particle encapsulates both a NN parameter set and computation (*e.g.*, code for forward and backward passes). We observe that many BDL inference algorithms can be recast in terms of particles. In particular, many BDL algorithms require the ability to create arbitrary numbers of particles and support communication between particles to convey parameter updates. We walkthrough both extremes on the spectrum of communication patterns between NN particles in common BDL algorithms now.

```
1   def _gather(particle: Particle) -> None:
2       # 1. Determine other particles
3       other_particles = [pid for pid in particle.particle_ids()
4           if pid != particle.pid]
5       # 2. Gather every other particle's parameters
6       futures = {pid: particle.get(p) for pid in other_particles}
7       # 3. Wait for result
8       particles = {pid: future.wait() for pid, future in futures.items()}
9       # 4. View a particle's parameters
10      particles[other_particles[0]].view()
```

Figure 1: The function _gather demonstrates an all-to-all communication pattern between particles in Push that uses a blend of an actor-model and async-await style of concurrency (§ 3.2).

A deep ensemble (Lakshminarayanan et al., 2017) trains a fixed (and arbitrary) number of NNs independently of each other. Thus, deep ensembles require no communication between any of the particles. SWAG (Maddox et al., 2019) additionally computes the first and second moments of a NN's parameter trajectory during training. Multi-SWAG (Wilson & Izmailov, 2020) ensembles SWAG. SVGD (Liu & Wang, 2016) will train a fixed (and arbitrary) number of particles in a manner dependent on each other—at each step of SVGD, each particle will communicate with every other particle to determine its own parameter update. Thus, SVGD requires all-to-all communication between every particle.

The discussion above highlights some challenges of implementing BDL algorithms. Since each particle represents a NN, we may require extra effort to write code that deals with the increased compute requirements and memory usage that comes with increasing the number of particles. Additionally, BDL algorithms may require communication between particles. This may be tedious to implement without library support as well as produce different GPU utilization characteristics compared to typical training. Instead, we propose an abstraction that formalizes the notion of a particle in Push, enabling us to easily create particles so that we can seamlessly scale the number of particles and natively supports communication between particles so that information can be easily exchanged during Bayesian inference. In this way, we can more easily experiment with BDL algorithms without worrying about the low-level details of efficiently implementing the algorithms on multi-GPU systems.

## 3.2 PARTICLE ABSTRACTION

The design of Push is inspired by existing models of concurrency such as the actor model of concurrency (Hewitt et al., 1973). In the actor model, an *actor* is a basic building block that has local state, can perform local computations, create more actors, send messages to other actors, and respond to messages from other actors. Similarly, a *particle* in Push is a basic building block that has local state (*e.g.*, NN parameters), can perform local computations (*e.g.*, forward passes and backward passes), create more particles (*e.g.*, replicate a NN), send messages to other particles (*e.g.*, trigger a computation), and respond to messages from other particles (*e.g.*, send my parameters to another particle). While we take inspiration from existing models of concurrency, we emphasize that the above is an analogy and Push's focus on BDL suggests different design decisions and interface.

A particle wraps a NN with local state, its own logical thread of execution (*i.e.*, its own control-flow), and message-passing capabilities so that each particle can execute concurrently alongside other particles and asynchronously exchange information. Each particle has a dictionary that associates messages with locally defined functions that defines how it will respond to messages it receives. This dictionary is populated with a receive method (Figure 2) that takes a message msg and function fn as arguments, and indicates that a particle should perform fn when it receives a message msg. Each particle also has a unique identifier pid. This identifier can be referenced by others particles for the purposes of sending messages. Ordinary NN functionality such as forward and backward passes execute on its own logical thread of execution, separate from other particle computations. We walkthrough an excerpt of an all-to-all communication pattern in Push now to illustrate the message-passing semantics by example (Figure 1).

```
1  def push_all_to_all_main(n: int, nn: torch.nn.Module, *args) -> None:
2      with PusH(nn, *args) as pd:
3          # 1. Create n particles that can gather
4          pids = [pd.p_create(mk_empty_optim,
5                              receive={"GATHER": _gather}) for p in range(n)]
6          # 2. Launch particle 0 and wait
7          pd.p_wait([pd.p_launch(0, "GATHER")])
```

Figure 2: The function `push_all_to_all_main` uses particles to define a Push distribution.

The function `_gather` implements an all-to-all communication pattern between all particles (Figure 1). On Line 3, the code `particle.particle_ids()` gives each particle access to particle identifiers for every other particle that is present where `particle` gives the currently executing particle access to its own local state. On Line 6, the code `particle.get(pid)` asynchronously accesses the state of particle `pid`. The method `get` is a special case of a more general `send` method that asynchronously sends a message `msg` to particle `pid` with optional arguments `*args` to trigger the function associated with `msg` on the receiving particle. The receiving particle executes the function associated with `msg` to the arguments `*args` on its own timeline and returns the result to the calling particle when it is finished. In the meantime, the calling particle receives a *future* (type `PFuture`) from the receiving particle so that the caller can optionally wait on the receiver to finish the computation. On Line 8, the code `future.wait()` blocks execution of the particle until the future has resolved. Thus, Push blends contains concepts from both actor-based concurrency and async-await concurrency (Syme et al., 2011). Finally, on Line 10, we `view` the result to obtain a read-only copy of a particle's parameters.

### 3.3 MODELS IN PUSH

Push users define a model by defining a *Push distribution*, abbreviated PD. A PD, written $\mathcal{P}(nn_\Theta)$, is parameterized by an input NN architecture $nn_\Theta$ with parameters $\theta \in \Theta$. Thus, Push users specify a model directly with a NN architecture as opposed to a generative process as in traditional probabilistic programming approaches. A PD uses the input NN as a template to create particles, and thus, encapsulates a set of particles $\{nn_{\theta_1}, \ldots, nn_{\theta_n}\}$ where $nn_{\theta_i}$ instantiates the NN architecture $nn_\Theta$ with parameters $\theta_i \in \Theta$. A PD approximates a probability distribution

$$\mathcal{P}(nn_\Theta) = \frac{1}{n} \sum_{i=1}^{n} \delta_{nn_{\theta_i}}(nn_\Theta) \approx p(nn_\Theta) \qquad (1)$$

with particles where $p(nn_\Theta)$ defines a distribution on a fixed NN architecture with varying parameters $nn_\theta$ by defining a distribution on its parameters $\theta$, *i.e.*, a *Bayesian NN* (MacKay, 1995; Goan & Fookes, 2020; Izmailov et al., 2021; Jospin et al., 2022). We discuss the details of this approximation, which views a *particle as performing a pushforward* (hence Push's name), in Appendix A. A full characterization of the models definable in Push and the relation to Gaussian processes (*i.e.*, another distribution on functions) is beyond the scope of the current paper, and would be an interesting direction of future work.

The properties of the approximation (*e.g.*, smoothness) depends on the interaction between particles encapsulated in a PD. As a reminder, interactions between particles can also be used to express BDL algorithms (recall § 3.1). This highlights that particles are core to both modeling and inference in Push. For this reason, we say that Push exposes model and inference at the same level of abstraction. Consequently, the quality of approximation of a $\mathcal{P}(nn_\Theta)$ can alternatively be viewed in light of assessing the quality of particle-based posterior inference algorithms targeting $p(\mathbf{Y}|nn_\Theta, \mathbf{X})\mathcal{P}(nn_\Theta)$ for $\mathcal{D} = (\mathbf{X}, \mathbf{Y})$.

Figure 2 illustrates how to use particles to define a a Push distribution (PD). On Line 2, we construct a PD `PusH` where `nn` is a generic PyTorch NN created with arguments `*args`. This wraps an ordinary NN in a form where the language implementation can now introspect and manipulate it to provide support for Bayesian inference. Line 4 initializes n particles using `p_create`. The particles are indexed by $0, \ldots, n-1$. Line 6 enables each particle to respond to `"GATHER"` messages from other particles with the `_gather` function. On Line 7, we asynchronously launch particle 0

(p_launch) and wait on the results (p_wait). Once a collection of particles has been defined, we can use message passing and local computations on particles to perform inference.

### 3.4 BAYESIAN DEEP LEARNING IN PUSH

Push users perform Bayesian inference on the model $p(\mathbf{Y}|nn_\Theta, \mathbf{X})\mathcal{P}(nn_\Theta)$ by expressing a BDL algorithm in terms of computations involving particles in $\mathcal{P}(nn_\Theta)$. A PD $\mathcal{P}(nn_\Theta)$ with particles $\{nn_{\theta_1}, \ldots, nn_{\theta_n}\}$ whose parameters have been updated with respect to data $\mathcal{D}$ can be used to approximate a posterior distribution on NN parameters with architecture $nn_\Theta$ as $p(\hat{nn}_\Theta|\mathcal{D}) \approx \frac{1}{n}\sum_{i=1}^{n} \delta_{nn_{\theta_i}}(\hat{nn}_\Theta)$. The expectation of $p(\hat{nn}_\Theta|\mathcal{D})$ is a function and can be approximated using this representation as the function $\hat{f}(x) = \frac{1}{n}\sum_{i=1}^{n} nn_{\theta_i}(x)$ that averages the predictions of each particle.

In general, a PD $\mathcal{P}(nn_\Theta)$ does not have a density. Thus, we cannot directly use Bayesian inference techniques that involve densities to compute the posterior $p(nn_\Theta|\mathcal{D})$ such as MCMC. There are at least two approaches to dealing with this issue. First, we can make additional assumptions about the posterior $p(nn_\Theta|\mathcal{D})$. For example, SWAG makes the assumption that the posterior distribution is a Normal distribution with mean and covariance dependent on the training trajectory.[1] Assumptions that introduce densities opens the possibility to apply more inference algorithms. Second, we can rely on score-based inference algorithms that depend on $\nabla_\theta \log p(nn_\Theta|\mathcal{D}) = \nabla_\theta \log p(\mathbf{Y}|nn_\Theta, \mathbf{X}) + \nabla_\theta \log \mathcal{P}(nn_\Theta)$ (see Appendix B). Note that a PD does not contain stochastic nodes so that the reparameterization trick does not need to be applied when back-propagating (Kingma & Welling, 2013)

We have implemented deep ensembles, SWAG, and SVGD in Push using particles to demonstrate its expressivity. We give an example SVGD implementation in Appendix B. Monte-Carlo dropout Gal & Ghahramani (2016) is directly implementable in the standard way. It is also possible to implement MCMC algorithms by using particles to represent proposals, possibly with gradients, and to implement message-passing style algorithms by defining the appropriate send functions to update other particles parameters with knowledge of the calling particle's parameters/state. We hope that the particle abstraction lowers the barrier to entry to experimenting with novel BDL algorithms. Algorithms such as Bayes-by-backprop Blundell et al. (2015) are not applicable in our representation since we do not directly define a distribution on parameters. Although our focus is on BDL, other learning algorithms might also benefit from the particle abstraction. For example, the particle abstraction may be useful for implementing metaheuristic algorithms that perform global optimization such as simulated annealing and genetic algorithms.

## 4 PUSH IMPLEMENTATION

Since models and inference algorithms are defined at the same level of abstraction in Push, the focus of the library is on providing an efficient implementation of the particle abstraction. This stands in contrast to many approaches to probabilistic programming where users define a model at a higher-level of abstraction compared to inference so that inference can be fully automated. We discuss the high-level architecture of Push (§ 4.1), key components of the implementation (§ 4.2 and § 4.3), and some implementation details (§ 4.4).

### 4.1 HIGH-LEVEL ARCHITECTURE

The primary challenge with implementing Push involves mapping the execution of particles to GPUs. To solve this problem, we introduce a separation of concerns between a PD, user-level particles, and the underlying hardware by adding a layer of indirection called a *node event loop* (§ 4.2). In this way, users can allocate and use particles without exact knowledge of the underlying hardware, relying on the system to handle these details. The idea to use a layer of indirection is common in distributed systems design, although the setting of BDL will suggest different design choices. Techniques for distributed NN training (Ben-Nun & Hoefler, 2019) are also related, al-

---

[1]Related algorithms such as SWA (Izmailov et al., 2018) and MultiSWAG (Wilson & Izmailov, 2020) are conceptually similar to SWAG.

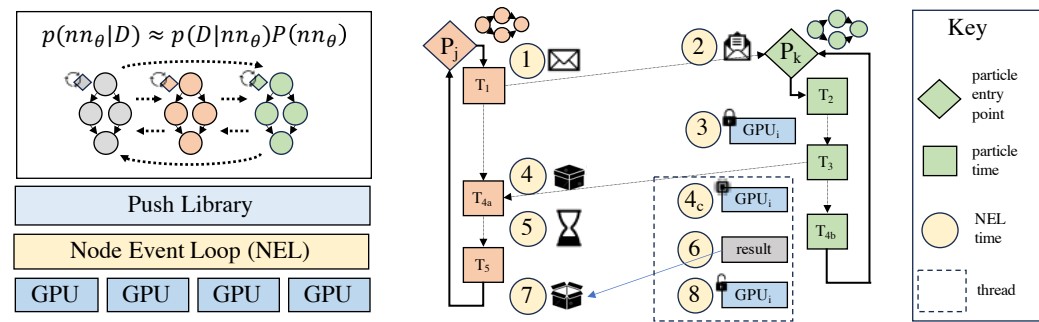

(a) Push provides a separation of concerns between particles and hardware.

(b) Timeline of events for two interacting particles in a NEL. Times $T_{4a}$, $T_{4b}$, and $4_c$ can execute concurrently.

Figure 3: High-level Push architecture.

though our challenges are orthogonal to those solved by data parallelism and model parallelism. Figure 3a illustrates the high-level architecture of Push, which we describe in more detail now.

## 4.2 NODE EVENT LOOP

A *node event loop* (NEL) provides the abstraction of an isolated process that handles all computations that utilize all the GPUs on a single physical compute node. A NEL is the core abstraction that enables the implementation of particles in Push. It is comprised of (1) a particle-to-device mapping and (2) a context switching mechanism. We discuss the components of a NEL in more detail now.

A NEL contains a particle-to-device lookup table to support all-to-all communication between particles. Each particle in a NEL is assigned to a unique device when it is created. Thus, different devices contain distinct subsets of particles. Note that a device can contain multiple particles since the user can define an arbitrary number of particles.

A NEL handles send/receive between particles/a PD. The particle-to-device lookup table is used to translate a particle to the appropriate device. If the particles involved in message passing are on different devices, data needs to be transferred to the appropriate device. This incurs communication overhead. If particles are on the same device, communication overhead can be eliminated. As send/receive messages are processed by a NEL, computations are dispatched to the GPU. The computations associated with send/receive include forward passes, backward passes, parameter updates, and user-defined functions. We discuss the mechanics of message processing next which involves context switching.

Figure 3b illustrates the mechanics of message processing by giving the timeline of events for two interacting particles that is supported by a NEL. A NEL maintains a single active particle that is computed on at any point in time while providing the illusion of concurrently executing particles to the user. In the example, $P_j$ is the active particle that the NEL is currently executing. $P_j$ sends a message to $P_k$ (Label 1, Figure 3b and Label 2, Figure 3b). To process this message, the NEL checks to see if the device the particle $P_k$ is mapped to is available (Label 3, Figure 3b). If it is available, the device is locked (Label 3, Figure 3b). If it is not available, the NEL will wait until the device is freed. Once the NEL has acquired a device, it will perform a *context switch*.

The context-switching dispatch mechanism enables a NEL to help manage GPU resource allocation and execute computations on different particles. In a traditional operating system, abstractions such as virtual memory and context-switching are used to share the compute and memory resources for threads/processes. Inspired by this, Push also contains a simple context-switching dispatch mechanism for particles. For each accelerator, the NEL maintains an *active set* which limits the maximum number of active particles allowable on a single device. This number is adjustable by the user. Particles in the active set exist in GPU memory and are pinned in a *particle cache*. Particles in the NEL that are not in the active set are stored off the accelerator to be swapped in when needed. We use the NEL's call stack to switch the active particle to the receiving particle $P_k$ to begin executing code in it's local execution context (*i.e.*, stack frame).

Once a context switch has successfully occurred, 3 things occur. First, the NEL will launch a thread to dispatch NN computations (forward passes, backward passes, parameter updates) to the GPU (Time 4c, Figure 3b). This thread of computation is what enables a NEL to potentially provide speedup since multiple threads can be executed concurrently on multiple devices. Second, the NEL will wrap the eventual results of computation in a future (Time 4a, Figure 3b). Third, the NEL will transfer control back to particle $P_j$ from $P_k$, making $P_j$ the active particle again and effectively blocking execution of $P_k$ until a further message is received (Time 4b, Figure 3b). At some point in the future, the thread that was launched will terminate (Label 6, Figure 3b). This may occur if $P_j$ waits on the future (Label 5, Figure 3b) or because the result has already been computed. At this point, the device will be freed to handle more computations (Label 8, Figure 3b). $P_j$ will have access to the result of the computation (Label 7, Figure 3b), which it can use to update its state or perform further computations, before blocking to receive a message.

Message processing on a NEL attempts to maximize concurrent usage of device accelerators while minimizing the amount of device contention. An alternative design to a NEL would be to wrap each particle in a lightweight process that might share a physical GPU. This design is similar to approaches taken by concurrent programming languages such as Erlang. We choose to execute all computations on a single NEL that launches threads because we prefer performance over fault-tolerance, *i.e.*, robustness to failure. In particular, communication with an accelerator device such as a GPU is already asynchronous. Consequently, introducing more processes may incur more overhead. The tradeoff is that if the NEL fails, than the entire system fails.

### 4.3 PUSH DISTRIBUTION

A PD executes in a separate process from a NEL. When a PD is created, it will create a NEL. Upon finishing all PD computations, the PD implementation will cleanup and exit the NEL. Currently, a PD does not support distributed NELs although there is no technical reason why Push cannot since NELs are isolated processes with point-to-point communication. It would be an interesting direction of future work to explore a distributed Push implementation.

### 4.4 IMPLEMENTATION DETAILS

Push is implemented in Python on top of PyTorch (Paszke et al., 2019). This enables us to take advantage of an established ecosystem and interoperate with a popular deep learning framework. We use Python threading to implement the concurrent semantics of a NEL and multiprocessing to isolate a NEL from a PD. This architecture enables us to extend a NEL to a distributed setting in future work.

## 5 EXPERIMENTS

We evaluate Push in two ways. First, we study the scaling of particles in Push across architectures, tasks, and methods to explore Push's expressivity and performance under different workloads (§ 5.1). Second, we study the scaling of Push as we tradeoff NN depth (*i.e.*, size) versus number of particles to compare with a traditional setting (*i.e.*, 1 particle) (§ 5.2).

### 5.1 SCALING OF PARTICLES ACROSS ARCHITECTURES, TASKS, AND METHODS

Figure 4 reports the scaling of particles in Push across devices (1, 2, and 4 GPUs) for selected architectures and tasks. We select three architectures and tasks to test a variety of workloads and Push's expressivity. We are able to apply Push to a variety of architectures and tasks with minimal code changes. We report more architectures and tasks in the Appendix C.

The first architecture we select is a Vision transformer (Dosovitskiy et al., 2021) applied to an image classification task on the MNIST dataset (LeCun, 1998). This is a standard task with a popular architecture. The second architecture we select is a convolutional NN applied to graphs called CGCNN (Xie & Grossman, 2018; Chanussot* et al., 2021) applied to a regression task on the MD17 dataset (Chmiela et al., 2017). CGCNN is a network specifically designed to model properties of atomistic systems. Thus, it is a domain specific architecture. The training of CGCNN to fit a potential energy surface will involve second-order derivatives. The third architecture we select is

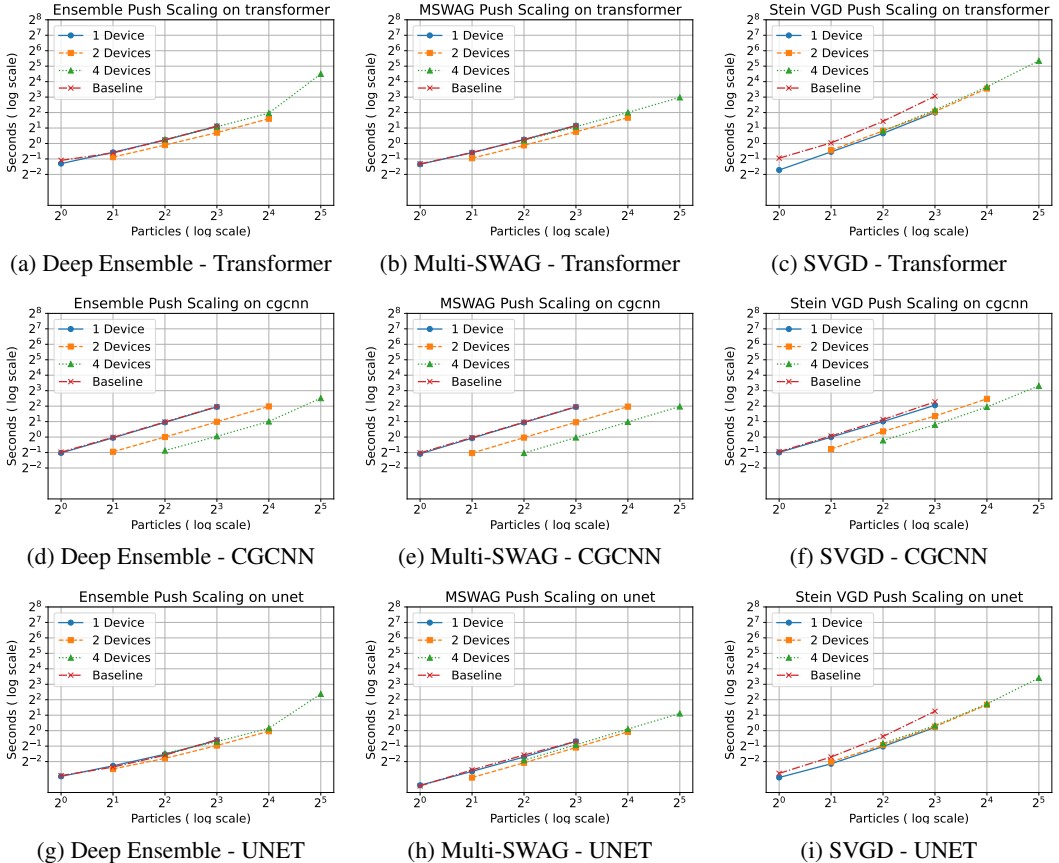

Figure 4: Scaling of particles in Push across devices (1, 2, and 4 GPUs) for selected architectures and tasks. The reported time per epoch is averaged across 10 epochs.

Unet (Ronneberger et al., 2015). Unet is a convolutional NN that is applied to learn a partial differential equation on the Advection dataset (Takamoto et al., 2023). Uncertainty quantification (Zhu & Zabaras, 2018; Zhu et al., 2019; Yang et al., 2021; Psaros et al., 2023) is important in these contexts since scientists and engineers want to provide guarantees on the trustworthiness of surrogate models and inferred solutions to inverse problems.

For each network, we test deep ensembles, multi-SWAG, and SVGD. We also test for 1 device $\{1, 2, 4, 8\}$ particles, for 2 devices $\{2, 4, 8, 16\}$ particles, and for 4 devices $\{4, 8, 16, 32\}$ particles. We report the number of particles and the number of seconds in log-scale. For the purposes of comparing the scaling of particles in Push across a diverse range of tasks, we select 40 batches of data to test in a single epoch. We use a batch size of 128 for the vision task, a batch size of 20 for CGCNN (consistent with training quantum chemistry networks (Chanussot* et al., 2021)), and a batch size of 50 for Unet (consistent with PDEBench (Takamoto et al., 2023)).

The blue curve shows the result for 1 device, the orange curve shows the result for 2 devices, and the green curve show the result for 4 devices. We make several observations. First, deep ensembles have the best scaling, since we can double the number of particles while holding the time per epoch constant by doubling the number of devices in Push. Second, SVGD has the worst scaling, since the computation of a kernel matrix requires all-to-all communication. We obtain benefits at lower particle counts. At higher particle counts, the SVGD algorithm is fundamentally bottlenecked by the computation of the kernel matrix. We do achieve benefits in the case of CGCNN using Push since higher-order derivatives and the graph stucture require more computation per particle. Third, multi-SWAG also scales since it essentially augments a deep ensemble with more particle-independent computation and is not fundamentally limited by communication patterns.

| Configuration | | 1 Device | | 2 Devices | | 4 Devices | |
|---|---|---|---|---|---|---|---|
| Parameters | D | P | Time (sec.) | P | Time (sec.) | P | Time (sec.) |
| 454089994 | 64 | 1 | $T_1$ | 2 | $\approx 1.04 \times T_1$ | 4 | $\approx 1.45 \times T_1$ |
| 227278090 | 32 | 2 | $T_2 \approx 1 \times T_1$ | 4 | $\approx 1.04 \times T_2$ | 8 | $\approx 1.29 \times T_2$ |
| 113872138 | 16 | 4 | $T_3 \approx 1 \times T_1$ | 8 | $\approx 1.04 \times T_3$ | 16 | $\approx 1.3 \times T_3$ |
| 57169162 | 8 | 8 | $T_4 \approx 1 \times T_1$ | 16 | $\approx 1.02 \times T_4$ | 32 | $\approx 1.46 \times T_4$ |
| 28817674 | 4 | 16 | $T_5 \approx 1.04 \times T_1$ | 32 | $\approx 1.09 \times T_5$ | 64 | $\approx 1.54 \times T_5$ |
| 14641930 | 2 | 32 | $T_6 \approx 1.1 \times T_1$ | 64 | $\approx 1.04 \times T_6$ | 128 | $\approx 1.77 \times T_6$ |
| 7554058 | 1 | 64 | $T_7 \approx 1.22 \times T_1$ | 128 | $\approx 1.14 \times T_7$ | 256 | $\approx 2.17 \times T_7$ |

Table 1: Depth (D) versus number of particles (P) tradeoff across devices. When we double device count, we double the effective parameter count. Ideal scaling has $1\times$ multiple (for 2 and 4 devices).

We also include baseline (*i.e.*, handwritten) implementations for comparison. On 1 device, we see that the performance of Push on 1 device for deep ensembles and multi-SWAG matches that of the baseline implementations. Thus, the overhead that Push introduces is minimal for 1 device. There are differences between the baseline implementation of SVGD and Push's implementation. In our baseline implementation, we store the kernel matrix and then update all the parameters after the kernel matrix has been computed since we only keep one copy of each NN. In Push's implementation, we use message passing between particles which provides read-only copies of particle parameters so that particle parameters can be updated concurrently. Thus, we see that Push's 1 device performance exceeds that of the baseline implementation.

## 5.2 SCALING: DEPTH VERSUS NUMBER OF PARTICLES

In practice, we may fix our compute budget so that we can only use a fixed *effective parameter count*, *i.e.*, size of each particle times the number of particles. As a result, it will be crucial to consider the tradeoff between the size of each particle (*e.g.*, its depth) and the number of particles. Table 1 reports the scaling behavior of Multi-SWAG in Push across devices when we divide the effective parameter count between the number of particles and the depth of each particle.

We select the vision transformer architecture and apply the multi-SWAG algorithm. We hold the effective parameter count constant by halving the size of the transformer (halving number of layers, $\{64, 32, 16, 8, 4, 2, 1\}$) while doubling the number of particles. We report the average epoch time across 5 epochs on 40 batches with batch size of 128. We test 1 device, 2 devices, and 4 device architectures. When we double the device count, we double the effective parameter count. Consequently, the 4 device configuration has 4 times the effective parameter count of a 1 device setting.

We observe two trends. First, our implementation supports larger models more efficiently as opposed to smaller models. In each column, we hold the effective parameter size constant but observe increasing overhead. This is a function of how efficiently the underlying NN library can utilize the GPU. Second, there is less overhead for scaling across devices when larger parameter counts are involved. As a reminder, as we double the number of devices, we double the effective parameter count. Thus, the 2 device times and 4 device times would ideally be $1\times$ the corresponding 1 device times provided that context switching is "free". At 128 particles on 2 devices, we have roughly 14% overhead while at 256 particles on 4 devices, we have roughly 117% overhead.

We present further scaling experiments in Appendix C. We saturate Push's performance on 1024 particles (small NNs) on 4 devices as a stress test of Push's implementation (Appendix C.3). We also study the tradeoff between the number of particles and particle size in multi-SWAG's accuracy, and observe that it is possible to achieve higher accuracy with more particles on smaller models while holding the effective parameter count constant (Appendix C.4). We emphasize that our focus in this paper is on Push and its scaling properties, and present this as an example of the exploration of BDL algorithms that we hope Push can enable.

## 6 DISCUSSION

We explore the combination of probabilistic programming, NNs, and concurrency in the context of BDL in this paper. In particular, we demonstrate (1) how particles can be used to approximately describe distributions on NNs in Push and (2) BDL algorithms can be implemented as concurrent procedures on particles that interact seamlessly with gradients. We also explore the challenges of scaling such an implementation and demonstrate promising single-node multi-GPU scaling results.

Extending Push to a distributed setting is an interesting direction of future work. Another interesting direction is to extend Push so that it supports a single particle across devices so that we can use larger models. Our hope is that Push can be used to more easily develop and test the performance of novel BDL algorithms that have more expressive communication patterns and at larger scale.

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

# A   MODELS IN PUSH

In the main text (Section 3.3), we claim that a PD approximates a probability distribution

$$\mathcal{P}(nn_\Theta) = \frac{1}{n} \sum_{i=1}^{n} \delta_{nn_{\theta_i}}(nn_\Theta) \approx p(nn_\Theta) \tag{2}$$

with particles where $p(nn_\Theta)$ defines a distribution on a fixed NN architecture with varying parameters $nn_\theta$ by defining a distribution on its parameters $\theta$, *i.e.*, a *Bayesian NN*. More formally,

$$p(nn_\Theta) \in \{p(nn_\theta) : \theta \in \Theta\} \cong \{p(\theta) : \theta \sim p(\theta), \theta \text{ parameters of } nn_\Theta\} \tag{3}$$

associates a distribution $p(nn_\theta)$ on a NN architecture $nn_\Theta$ from a class of NN architectures $\mathcal{NN}[\Theta]$ with a distribution $p(\theta)$ on the NN $nn_\theta$'s parameters. We emphasize the NN architecture is fixed and we only vary the parameters. We will unpack the claim now.

Consider a NN $nn(\cdot; \theta) : X \to Y \in \mathcal{NN}[\Theta]$. We can define a distribution on a parameterized family of distributions $\{nn(\cdot; \theta) : \theta \in \Theta\}$ by defining a distribution on its parameters using the correspondence in Equation 3. For example,

$$\mathbf{nn}(\cdot; \theta) \text{ where } \theta \sim \mu \tag{4}$$

defines a random function. We can approximate the distribution $\mu$ with independent and identically distributed (i.i.d.) samples $\{\theta_1, \ldots, \theta_n\}$ to create an empirical distribution $\mu_n^\delta$. Then

$$\mathbf{nn}(\cdot; \theta^\delta) \text{ where } \theta^\delta \sim \mu_n^\delta \tag{5}$$

also defines a random function. Intuitively, we have that

$$\mathbf{nn}(\cdot; \theta^\delta) \approx \mathbf{nn}(\cdot; \theta) \tag{6}$$

where the approximation becomes exact as the number of samples goes to infinity since $\mu_n^\delta \to \mu$ in distribution by the Glivenko–Cantelli theorem. Observe that the samples $\{\theta_1, \ldots, \theta_n\}$ are exactly particles of a PD $\mathcal{P}(nn_\theta)$. Thus, we have that

$$\mathcal{P}(nn_\theta) = \mathbf{nn}(\cdot; \theta^\delta) \approx \mathbf{nn}(\cdot; \theta). \tag{7}$$

The distribution of $\mathbf{nn}(x; \theta)$ for each $x$ can be directly given in terms of the pushforward of $\mu$ w.r.t. $nn_\theta$. As a reminder, the pushforward of a distribution $\mu$ on $\Theta$ with respect to (w.r.t.) $F : \Theta \to Y$ is $\mu_*(F) = B \mapsto \mu(F^{-1}(B))$ for any measurable subset $B$ of $\Theta$. The pushforward can be approximated as an empirical pushforward

$$\mu_*^\delta(F) = \{F(\theta_1^\mu), \ldots, F(\theta_n^\mu)\} \tag{8}$$

which approximates $\mu_*(F)$ using samples $\{\theta_1^\mu, \ldots, \theta_n^\mu\}$ where the superscript indicates that the particles are chosen in a manner dependent on $\mu$ and will be dropped from now on to declutter the notation. We have that $\mu_*^\delta(F) \to \mu_*(F)$ in distribution as well. We introduce the *particle pushforward* now.

**Definition 1** (particle pushforward). *Let $h(x; \theta)$ be a parameterized (measurable) function.*

*Define*

$$ppush(\mu)(h(x; \cdot)) = \mu_*(h(x; \cdot)) \tag{9}$$

*when the parameter $\theta$ is random (i.e., the argument $x$ is given).*

**Definition 2** (Empirical particle pushfoward). *Similarly, we define*

$$ppush^\delta(\mu)(h(x; \cdot)) = \mu_*^\delta(h(x; \cdot)) \tag{10}$$

*for the empirical version.*

Using these definitions, we have that

$$\mathbf{nn}(x; \theta) \sim ppush(\mu)(nn(x; \cdot)) \tag{11}$$

and

$$\mathbf{nn}(x; \theta^\delta) \sim ppush^\delta(\mu)(nn(x; \cdot)) \tag{12}$$

since the particle pushforward unfolds to defining a random variable $\theta$ with distribution $\mu$. Hence, we name Push after the **p**article **push**forward since it pushes the distribution on parameters along each particle using the structure of the NN. The method `p_create` creates a particle via `ppush`.

## A.1 A Compositional Perspective

In practice, we define NNs by composing smaller NN modules to build larger NN modules. We provide a more detailed analysis of the particle pushforward in this section to address this common case. The main reason for doing so is to illuminate the compositional structure underlying the particle pushforward that can be taken advantage of in inference algorithms and future implementations of Push. For example, we will see that the the particle pushforward is compositional which means future implementations and algorithms can replicate only part of a particle for efficiency purposes. We walk through a two-layer NN before considering arbitrary layers.

Consider a NN $nn(\cdot; \theta) : X \to Y$ defined as

$$nn(x; \theta^1, \theta^2) = g(f(x; \theta^1); \theta^2) \tag{13}$$

where we have explicitly indicated the parameters $\theta = (\theta^1, \theta^2)$, $\theta^1 \in \Theta_1$, and $\theta^2 \in \Theta_2$. It is comprised of (measurable) functions $f(\cdot; \theta^1) : X \to Z$ and $g(\cdot; \theta^2) : Z \to Y$. This defines a family of parameterized functions

$$\mathcal{F}[\Theta] = \{g(f(x; \theta^1); \theta^2) : \theta_1 \in \Theta_1, \theta^2 \in \Theta_2\} \subset \mathcal{NN}[\Theta] \tag{14}$$

where we can select a different function by varying the parameters $\theta^1 \in \Theta_1$ and $\theta^2 \in \Theta_2$.

We can define a distribution on a family of parameterized functions $\mathcal{F}[\Theta]$ by defining distributions on a parameterized function's parameters using the correspondence in Equation 3. For example,

$$\mathbf{nn}(x; \theta_\mathbf{1}, \theta_\mathbf{2}) = g(f(x; \theta_\mathbf{1}); \theta_\mathbf{2}) \tag{15}$$

where random variables $\theta_\mathbf{1} \sim \mu_1$ and $\theta_\mathbf{2} \sim \mu_2$ for some distributions $\mu_1$ and $\mu_2$. This gives a generative description of a distribution on $\mathcal{F}[\Theta]$. As before, the distribution

$$\mathbf{nn}(x; \theta_\mathbf{1}^\delta, \theta_\mathbf{2}^\delta) = g(f(x; \theta_\mathbf{1}^\delta); \theta_\mathbf{2}^\delta) \tag{16}$$

where random variables $\theta_\mathbf{1}^\delta$ and $\theta_\mathbf{2}^\delta$ are drawn from the empirical distributions of $\mu_1$ and $\mu_2$ is an approximation.

We extend the definitions above to work with product distributions, written $\mu \otimes \nu$. Define $(\mu \otimes \nu)_*^\delta(F) = \{(\theta_1^\mu, \theta_1^\nu) \mapsto F(\theta_1^\mu, \theta_1^\nu), \ldots, (\theta_n^\mu, \theta_n^\nu) \mapsto F(\theta_n^\mu, \theta_n^\nu)\}$ with particles $\theta_i^\mu$ and $\theta_i^\nu$ sampled from the distributions $\mu$ and $\nu$, respectively.

**Definition 3** (particle pushforward product). *Let $h(x; \theta)$ be a parameterized function. Define*

$$ppush(\mu_x \otimes \mu_\theta)(h) = (\mu_x \otimes \mu_\theta)_*(h) \tag{17}$$

*when both the argument and the parameter are random so that the first component defines the distribution on the argument and the second component defines the distribution on the parameter. Define*

$$ppush^\delta(\mu_x \otimes \mu_\theta)(h) = (\mu_x \otimes \mu_\theta)_*^\delta(h) \tag{18}$$

*similarly.*

Putting this together, we have that

$$\mathrm{ppush}^\delta(\mathrm{ppush}^\delta(\mu_1)(f(x; \cdot)) \otimes \mu_2)(g) \tag{19}$$

$$= \{(\theta_1^{\mu_1}, \theta_1^{\mu_2}) \mapsto g(f(x; \theta_1^{\mu_1}); \theta_1^{\mu_2}), \ldots, (\theta_n^{\mu_1}, \theta_n^{\mu_2}) \mapsto g(f(x; \theta_n^{\mu_1}); \theta_n^{\mu_2})\} \tag{20}$$

$$= \mathbf{nn}(x; \theta_\mathbf{1}^\delta, \theta_\mathbf{2}^\delta) \tag{21}$$

for any $x$. The particle pushforward thus unfolds to specifying a distribution on NNs with a generative approach using empirical distributions of $\mu_1$ and $\mu_2$. Since $\mu_1$ and $\mu_2$ are independent (by construction), and the empirical pushforward converges to the true distribution, we have defined the same distribution as if we had taken a generative approach.

Moreover, we also have

$$\mathcal{P}(nn(x; \theta^{\mu_1}, \theta^{\mu_2})) = \{(\theta_1^{\mu_1}, \theta_1^{\mu_2}) \mapsto g(f(x; \theta_1^{\mu_1}); \theta_1^{\mu_2}), \ldots, (\theta_n^{\mu_1}, \theta_n^{\mu_2}) \mapsto g(f(x; \theta_n^{\mu_1}); \theta_n^{\mu_2})\} \tag{22}$$

when the PD encapsulates particles $\{(\theta_1^{\mu_1}, \theta_1^{\mu_2}), \ldots, (\theta_n^{\mu_1}, \theta_n^{\mu_2})\}$. A PD is thus equivalent to $nn(x; \theta_\mathbf{1}^\delta, \theta_\mathbf{2}^\delta)$, completing the correspondence. We gather this into a lemma.

**Lemma 1.** *For any two-layer NN $nn(\cdot; \theta^{\mu_1}, \theta^{\mu_2})$,*

$$\mathcal{P}(nn(\cdot; \theta^{\mu_1}, \theta^{\mu_2})) = nn(x; \theta_1^\delta, \theta_2^\delta) \tag{23}$$

*where $\theta_1^\delta$ and $\theta_2^\delta$ are distributed according to the empirical distribution of $\mu_1$ and $\mu_2$.*

*Proof.* Unfolding definitions as above. $\square$

We can extend the reasoning above to $n$-layer feed-forward NNs by a suitable induction on $n$. More formally, define

$$\mathcal{F}_2[\Theta] = \{g(f(x; \theta_1); \theta_2) : \theta_1 \in \Theta_1, \theta_2 \in \Theta_2\} \tag{24}$$
$$\mathcal{F}_n[\Theta] = \{g(f(x; \theta_1); \theta_2) : f \in \mathcal{F}_{n-1}[\Theta_1], \theta_2 \in \Theta_2\} \tag{25}$$

when $n > 2$. The base case is the two-layer case which has already been shown. In the inductive case, we have by the inductive hypothesis that the empirical distribution has been defined for $\mu_1$ since $f \in \mathcal{F}[\Theta_1]$. The empirical distribution has been defined for $\mu_2$ as in the two-layer case.

## B  INFERENCE IN PUSH

We use the particle abstraction in Push to implement inference algorithms. As a reminder, Push exposes models and inference at the same level of abstraction via particles so that the user has flexibility to implement inference algorithms using the particle abstraction. The inference algorithm need only be written once and can be applied to multiple models. We provide more details in this section concerning how one might use the particle abstraction to encode inference algorithms. We also give the code for our implementation of SVGD in this section.

### B.1  SCORE-BASED INFERENCE ALGORITHMS

As a reminder, a PD does not have a density in general. In the main text, we mention that one method around this is to use a score-based inference algorithm (*e.g.*, SVGD) that uses an update of the form

$$\nabla_\theta \log p(nn_\Theta | \mathcal{D}) = \nabla_\theta \log p(\mathbf{Y} | nn_\Theta, \mathbf{X}) + \nabla_\theta \log \mathcal{P}(nn_\Theta). \tag{26}$$

We unpack this in more detail now, starting with the second term.

While $\mathcal{P}(nn_\Theta)$ may not have a density, we know from Appendix A that there is a corresponding $p(\theta)$ associated with each $\mathcal{P}(nn_\Theta)$ as the number of particles tends to infinity. Thus, we can approximate

$$\nabla_\theta \log p(nn_\Theta | \mathcal{D}) \approx \nabla_\theta \log p(\mathbf{Y} | nn_\Theta, \mathbf{X}) + \nabla_\theta \log p(\theta). \tag{27}$$

The term $\nabla_\theta \log p(\theta)$ is tractable if $p(\theta)$ has a density that is differentiable.

The first term $\nabla_\theta \log p(\mathbf{Y} | nn_\Theta, \mathbf{X})$ can be reduced to a backwards pass of a particle $nn_\Theta$ using an appropriate loss applied to $nn_\Theta$'s predictions. For example, suppose $p(\mathbf{Y} | nn_\Theta, \mathbf{X})$ is a Normal distribution centered at $nn_\Theta$'s predictions. In symbols,

$$\mathbf{Y} \sim \mathcal{N}(nn_\Theta(\mathbf{X}), I) \tag{28}$$

where $\mathcal{N}$ is a multivariate Normal distribution with mean $nn_\Theta(\mathbf{X})$ and identity covariance. In this case, we recover the familiar mean-squared error (MSE) loss since

$$\log p(\mathbf{Y} | nn_\Theta, \mathbf{X}) \propto (\mathbf{Y} - nn_\Theta(\mathbf{X}))^2. \tag{29}$$

The gradient $\nabla_\theta \log p(\mathbf{Y} | nn_\Theta, \mathbf{X})$ thus corresponds to a backwards pass of a NN w.r.t. $\theta$. We use this encoding to implement parts of the SVGD algorithm in Push. More generally, we can use this encoding to implement other scored-based inference algorithms in Push that exchange information between particles and use gradients in various ways.

```
1   class SteinVGD(Infer):
2       def __init__(self, mk_nn: Callable, *args: any,
3                    num_devices=1, cache_size=4, view_size=4) -> None:
4           super(SteinVGD, self).__init__(
5               mk_nn, *args, num_devices=num_devices, cache_size=cache_size, view_size=view_size)
6
7       def bayes_infer(self, dataloader: DataLoader, epochs: int,
8                       prior=None, loss_fn=torch.nn.MSELoss(),
9                       num_particles=1, lengthscale=1.0, lr=1e-3,
10                      svgd_entry=_svgd_leader, svgd_state={}):
11          pid_leader = self.push_dist.p_create(mk_empty_optim, device=0, receive={
12              "SVGD_LEADER": svgd_entry
13          }, state=svgd_state)
14          pids = [pid_leader]
15          for p in range(num_particles - 1):
16              pid = self.push_dist.p_create(
17                  mk_empty_optim, device=((p + 1) % self.num_devices),
18                  receive={
19                      "SVGD_STEP": _svgd_step,
20                      "SVGD_FOLLOW": _svgd_follow
21                  },
22                  state=svgd_state)
23              pids += [pid]
24          self.push_dist.p_wait([self.push_dist.p_launch(0, "SVGD_LEADER",
25              prior, loss_fn, lengthscale, lr, dataloader, epochs)])
```

Figure 5: Top-level SVGD class in Push.

## B.2 STEIN VARIATIONAL GRADIENT DESCENT

Figure 5 gives the top-level code for SVGD defined in Push using the particle abstraction. We can define new inference methods in Push by extending the `Infer` class. On Lines 4-5, we create a PD object with the appropriate NN. The number of devices `num_devices`, cache size `cache_size`, and view size (`view_size`) for viewing other particle's parameters can be set by the user. These parameters give the user control over how a compute node's accelerator devices are used. In general, we would like to set the cache size and view size as large as possible to minimize context switching, with an upper bound being the number of particles.

Lines 7 to 25 set up the particles in a PD. One line 11 through 13, we create a leader particle `pid_leader` that serves as a synchronization point between every other particle. It responds to `"SVGD_LEADER"` messages with the function `svgd_entry` (Figure 6). We create other particles on lines 15 to 20 that respond to `"SVGD_STEP"` and `"SVGD_FOLLOW"` messages with `_svgd_step` and `_svgd_follow` respectively (Figure 6). We describe these functions in turn now.

The function `_svgd_leader` performs the bulk of the computation. It first computes the gradient of every particle w.r.t. the current data point (Lines 10-12). We must wait on all gradients to finish computing before gathering all other particle parameters and their gradients on the leader particle (Lines 14-18). On Lines 20-43, we compute the SVGD kernel update for the RBF kernel and send an update to each particle. The functions `_svgd_step` and `_svgd_follow` step and update the parameters of the particles respectively.

We make several comments about the Push implementation of SVGD. First, note that we can only update a particle's parameters after we have computed the entire pairwise kernel matrix for the previous iteration's parameter. The view in Push provides a read-only copy of a particle's parameters so that we do not need to maintain a copy of parameters. Second, the implementation of SVGD is agnostic to the number of devices that are available to the language implementation. Consequently, we can seamlessly scale the number of devices without changing the implementation of SVGD. This holds more broadly for any inference algorithm written in Push. Third, the implementation of SVGD uses both actor-based and async-await concurrency. Each particle executes concurrently and particles can wait on other particles to finished computation. While this programming model is more complex than ordinary sequential computing, it offers the potential to scale to more devices and a distributed setting with minimal changes. Note that a Push inference algorithm need only be written once.

```
1   def _svgd_leader(particle: Particle, prior, loss_fn: Callable,
2                    lengthscale, lr, dataloader: DataLoader, epochs) -> None:
3       n = len(particle.particle_ids())
4       other_particles = list(filter(lambda x: x != particle.pid, particle.particle_ids()))
5
6       for e in tqdm(range(epochs)):
7           losses = []
8           for data, label in dataloader:
9               # 1. Step every particle
10              fut = particle.step(loss_fn, data, label)
11              futs = [particle.send(pid, "SVGD_STEP", loss_fn, data, label)
12                      for pid in other_particles]
13              fut.wait(); [f.wait() for f in futs]
14
15              # 2. Gather every other particles parameters
16              particles = {pid: (particle.get(pid) if pid != particle.pid else
17                  list(particle.module.parameters())) for pid in particle.particle_ids()}
18              for pid in other_particles:
19                  particles[pid] = particles[pid].wait()
20
21              def compute_update(pid1, params1):
22                  params1 = list(particles[pid1].view().parameters())
23                              if pid1 != particle.pid else params1
24                  p_i = flatten(params1)
25                  update = torch.zeros_like(p_i)
26                  for pid2, params2 in particles.items():
27                      params2 = list(particles[pid2].view().parameters())
28                                  if pid2 != particle.pid else params2
29
30                      # Compute kernel
31                      p_j = flatten(params2)
32                      p_j_grad = flatten([p.grad if p.grad is not None else torch.zeros_like(p)
33                                  for p in params2])
34                      diff = (p_j - p_i) / lengthscale; radius2 = torch.dot(diff, diff)
35                      k = torch.exp(-0.5 * radius2).item()
36                      diff.mul_(-k / lengthscale)
37                      update.add_(p_j_grad, alpha=k); update.add_(diff)
38                  update = update / n
39                  return update
40
41              # 3. Compute kernel and kernel gradients
42              for pid1, params1 in particles.items():
43                  # 4. Send kernel
44                  if pid1 != particle.pid:
45                      update = compute_update(pid1, params1)
46                      particle.send(pid, "SVGD_FOLLOW", lr, update).wait()
47              update = compute_update(particle.pid, particles[particle.pid])
48              _svgd_follow(particle, lr, update)
49              loss = loss_fn(particle.forward(data).wait().to("cpu"), label)
50              losses += [torch.mean(torch.tensor(loss))]
51
52  def _svgd_step(particle: Particle, loss_fn: Callable,
53                 data: torch.Tensor, label: torch.Tensor, *args:any) -> torch.Tensor:
54      return particle.step(loss_fn, data, label,*args)
55
56  def _svgd_follow(particle: Particle, lr: float, update: List[torch.Tensor]) -> None:
57      # 1. Unflatten
58      params = list(particle.module.parameters())
59      updates = unflatten_like(update.unsqueeze(0), params)
60
61      # 2. Apply the update to the input particle
62      with torch.no_grad():
63          for p, up in zip(params, updates):
64              p.add_(up.real, alpha=-lr)
```

Figure 6: SVGD in Push.

## C  Experiments

### C.1  More Experimental Details

**Hardware**  We run experiments 2xA5000 and 4xA5000 machines using Paperspace https://www.paperspace.com/. Each A5000 GPU has 24 GB of video ram each. The 2xA5000 machine uses a Intel(R) Xeon(R) Gold 5315Y CPU @ 3.20GHz that has 16 cores and 90 GB of

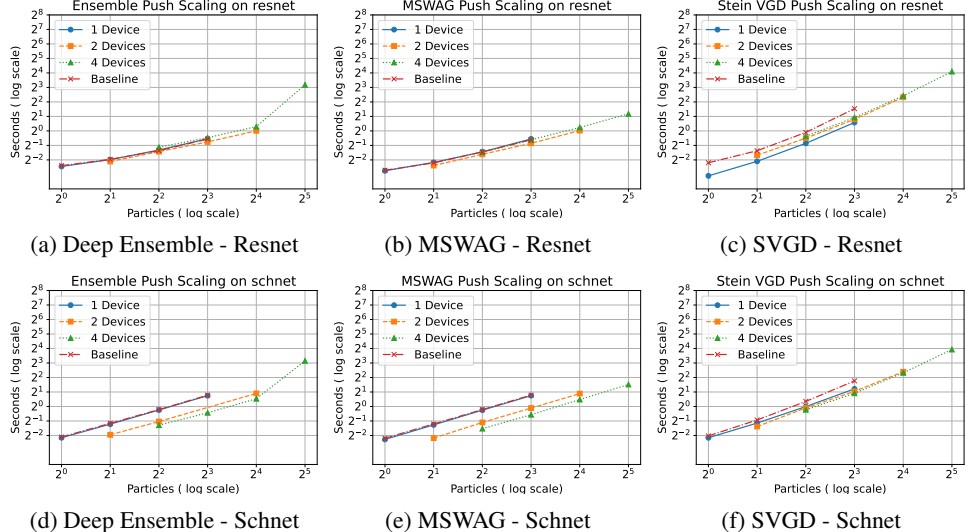

(a) Deep Ensemble - Resnet  (b) MSWAG - Resnet  (c) SVGD - Resnet

(d) Deep Ensemble - Schnet  (e) MSWAG - Schnet  (f) SVGD - Schnet

Figure 7: Scaling of particles in Push across devices (1, 2, and 4 GPUs) for selected architectures and tasks. The reported time per epoch is averaged across 10 epochs.

ram. The 4xA500 machine uses uses a Intel(R) Xeon(R) Gold 5315Y CPU @ 3.20GHz that has 32 cores and 180 GB of ram.

**Architecture details**    For our transformer experiments in Figure 4, we use the b16 vision transformer from the PyTorch vision library with an image size of 28, patch size of 14, 10 classes, 8 heads, 16 layers, MLP dimension of 1280, and hidden dimension of 320. We use this smaller transformer to ensure that we can compare across 1, 2, and 4 devices fairly.

For our transformer experiments in Table 1, we use the b16 vision transformer from the PyTorch vision library with an image size of 28, patch size of 14, 10 classes, 12 heads, varying number of layers (as specified in the table), MLP dimension of 3072, and hidden dimension of 768. Thus, we use the default settings with the exception of varying the number of layers.

For our CGCNN experiments, we use the default implementation give by the Open Catalyst Project (OCP) (Chanussot* et al., 2021) without modification. We note that the original CGCNN network was not original designed for the potential energy surface task that we applied it to in our experiments and that OCP also applies it to.

For our Unet experiments, we use the default implementation provided by PDEBench (Takamoto et al., 2023) without modification.

## C.2    ADDITIONAL SCALING EXPERIMENTS

Figure 7 contains additional architectures and tasks that we tested Push on to test its performance on a variety of workloads and its ability to handle a variety of NNs. These architectures include ResNet (He et al., 2016), a vision network, and Schnet (Schütt et al., 2017), a quantum chemistry network. The performance trends that we observe in these networks are consistent with what we observe in the main text. We make several more remarks to supplement the main text.

First, we reiterate again that the baseline SVGD implementation is not identical to the one implemented in Push. This is because we do not use the particle abstraction in the baseline implementation. In contrast, all inference methods in Push use the particle abstraction. Unlike SVGD, deep ensembles can be easily implemented without the particle abstraction. multi-SWAG presents an interesting design point since we can encode the moments as additional particles to take advantage of Push's design. In our current work, we opt to make the Push implementation as close to the baseline implementation. Thus, our current implementation of multi-SWAG pays all the costs of using the particle abstraction without gaining any of the benefits.

| Parameters | 1 Device | | 2 Devices | | 4 Devices | |
| | P | Time (sec.) | P | Time (sec.) | P | Time (sec.) |
|---|---|---|---|---|---|---|
| 57169162 | 8 | $T_1$ | 16 | $\approx 1.02 \times T_1$ | 32 | $\approx 1.47 \times T_1$ |
| 38282506 | 16 | $T_2 \approx 1.57 \times T_1$ | 32 | $\approx 1.04 \times T_2$ | 64 | $\approx 1.75 \times T_2$ |
| 14428810 | 32 | $T_3 \approx 1.88 \times T_1$ | 64 | $\approx 1.32 \times T_3$ | 128 | $\approx 3.16 \times T_3$ |
| 7330954 | 64 | $T_4 \approx 2.03 \times T_1$ | 128 | $\approx 1.37 \times T_4$ | 256 | $\approx 3.3 \times T_4$ |
| 4968586 | 128 | $T_5 \approx 3.68 \times T_1$ | 256 | $\approx 1.49 \times T_5$ | 512 | $\approx 3.65 \times T_5$ |
| 1896010 | 256 | $T_6 \approx 7.03 \times T_1$ | 512 | $\approx 1.53 \times T_6$ | 1024 | $\approx 3.81 \times T_6$ |

Table 2: "Width" versus number of particles (P) tradeoff across devices. When we double device count, we double the effective parameter count. Ideal scaling has $1\times$ multiple (for 2 and 4 devices).

| Parameters | Depth | Standard Accuracy | Particles | multi-SWAG accuracy |
|---|---|---|---|---|
| 227278090 | 32 | 97.70 | 1 | 97.78 |
| 113872138 | 16 | 98.23 | 2 | 97.93 |
| 57169162 | 8 | 98.08 | 4 | 98.13 |
| 28817674 | 4 | 98.03 | 8 | 98.28 |
| 14641930 | 2 | 97.95 | 16 | 98.36 |
| 7554058 | 1 | 97.63 | 32 | 98.28 |

Table 3: Number of layers (depth $D$) versus number of particles tradeoff compared to standard training.

Second, we do observe scaling issues across devices on small networks, even at small particle sizes, with Push. This is visible in a network like SchNet which is small. In general, all the performances on networks reported in Figure 4 and Figure 7 are a lower bound on performance, since all networks were chosen to be smaller so that we can fit 8 particles on a device. We note that libraries such as PyTorch are most effective when the amount of compute on the GPU outweighs the cost of communication with the GPU. Push inherits this property.

Third, Push provides the most advantage when the amount of computation per a network is high and when the amount of communication is low. It would be an interesting to further study the tradeoff between particle size and number of particles to determine the optimal implementation of Push and to ground experimentation with novel BDL algorithms.

## C.3 STRESS TEST

As a stress test of Push, we run up to 256 particles on 1 device, 512 particles on 2 devices, and 1024 particles on 4 devices. Table 2 reports the results. We use a similar experimental setup as in the main scaling experiment. However, since we cannot reduce the number of layers beyond 1, we instead choose to keep the number of layers at 12, and shrink the dimensions of the MLP and the hidden dimension in the transformer to keep the effective parameter count roughly constant as we increase the particle count. The performance of Push begins to saturate on 4 devices at 1024 particles, since it is $3.81\times$ slower than the 1 device version. Thus, while we can handle larger particle counts, we do not obtain speedup. The multi-device setup is still beneficial since swapping particles on and off the accelerator is even more costly. We believe the overheads can be improved with better scheduling algorithms.

## C.4 MULTI-SWAG ACCURACY

In the main text, we have focused on the scaling properties of Push on a variety of algorithms, architectures, and tasks. We do this because Push is a library that is agnostic to choice of algorithm, architecture, and task. However, in practice, we may want to know additional properties such as the effect on accuracy between standard learning and BDL when determining how to allocate a compute budget. In this section, we present some preliminary results on multi-SWAG to prepare for future work.

| Parameters | Standard accuracy | Particles | multi-SWAG accuracy |
|---|---|---|---|
| 170575114 | 97.89 | 1 | 97.81 |
| 85520650 | 97.70 | 2 | 97.56 |
| 57190666 | 97.94 | 4 | 98.18 |
| 21526666 | 97.76 | 8 | 98.31 |
| 14439562 | 97.85 | 16 | 98.18 |
| 5454922 | 97.22 | 32 | 97.87 |

Table 4: "Width" versus number of particles (P) tradeoff compared to standard training.

Table 3 gives the results of an experiment applying a vision transformer to image classification on the MNIST dataset. For the standard accuracy, we simply train a NN in the standard way for 10 epochs with the Adam optimizer and a learning rate of .001 For the multi-SWAG algorithm, pretrain each multi-SWAG particle for 7 epochs and perform multi-SWAG training for 3 epochs with the Adam optimizer and a learning rate of $0.001$.[2] To form a multi-SWAG prediction, we first draw 5 samples from each SWAG posterior for each particle with a variance of $1^{-30}$. We then predict the majority across all samples drawn from all particles. We decrease the parameter count by halving the depth. For standard training, we always use 1 particle so the parameter count is halved. For multi-SWAG, we double the number of particles so as to keep the effective parameter count constant. Thus, to compare the performance of multi-SWAG with standard training with the same effective parameter count, we should always compare with standard training on the largest model. We see evidence that multi-SWAG can outperform standard training at higher particle counts with smaller networks. Studying this tradeoff in more detail, and across more architectures and more training scenarios (*e.g.*, training longer) is an example of what we hope Push can enable.

Table 4 gives the results of another experiment applying a vision transformer to image classification on the MNIST dataset. We keep the experimental setup as before, the only difference being that we decrease the "width" of the network as opposed to the depth. We do this by decreasing the MLP size and the hidden dimension size of the transformer. As before, we see evidence that multi-SWAG can outperform standard training at higher particle counts with smaller networks.

---

[2]Vanilla stochastic gradient descent is recommended for multi-SWAG.

