# OpenReview forum: "Push: Concurrent Probabilistic Programming for Bayesian Deep Learning"
_ICLR.cc/2024/Conference — Submitted to ICLR 2024_

### Official Review · Reviewer_pwQF · 2023-10-30

**Soundness:** 3 good
**Presentation:** 2 fair
**Contribution:** 1 poor
**Rating:** 3
**Confidence:** 4

**Summary:**

The paper describes a library for distributed training of a particular form of Bayesian neural network. The library is based on PyTorch.

**Strengths:**

The paper describes a working library supporting Bayesian deep learning on multiple computing nodes. The paper is accompanied by the implementation's source code.

**Weaknesses:**

Despite the title and the abstract, it is not clear how the paper's contributions are related to either probabilistic programming or Bayesian inference. The library described in the paper implements a very basic communication protocol for distributed training, which is neigher novel nor specific to Bayesian machine learning. The protocol is applied to a couple of training algorithms, for which distributed execution is either trivial (Ensembe, SWAG) or was explored and implemented in prior work (SVGD).

The cited context of the paper is very broad, including general papers on probabilistic programming and  Bayesian statistical inference, however the paper does not  appropriately cite relevant work on Bayesian neural networks in probabilistic programming.  Contrary to what the paper states, Pyro, PyMC, and possibly other frameworks provide tools and tutorials for using BNNs in/as probabilistic programs, and training. Also, there is quite some work on describing inference and model on the same level in probabilistic programming, Gen being an example.  Publications accompany most of these innovations.

Empirical evaluation: the library described in the paper was only applied to a couple training algorithms, and evaluated on rather trivial benchmark problems. There are little insights that can be drawn from such limited evaluation.

**Questions:**

What does your library does better than Ray (https://www.ray.io/) for distributed training of Bayesian networks using SVGD?

---

> ### Author Response · Authors · 2023-11-16
>
> We thank you for your time and thoughtful feedback. We refer you to our general response for additional clarification and are happy to answer any more questions.
>
> > What does your library does better than Ray (https://www.ray.io/) for distributed training of Bayesian networks using SVGD?
>
> We thank the reviewer for this question. Our library currently focuses on enabling a representation of a Bayesian neural network as a communicating and concurrently executing ensemble of neural networks on single-node, multi-GPU systems. Thus, our library focuses on utilizing a single-node as efficiently as possible to efficiently implement a representation of a Bayesian neural network. Ray is a general purpose distributed training framework that can be used to execute neural network ensembles on multi-node and multi-GPU systems. Thus, Ray focuses on utilizing multiple-nodes as efficiently as possible in a setting where nodes may fail. It would be an interesting direction of future work to utilize Ray and our library to create a distributed and concurrent representation of a Bayesian neural network, and study inference algorithms on this representation.

---

> > ### Comment · Reviewer_pwQF · 2023-11-22
> >
> > I have read the authors' comments and I stand by my opinion about the paper.

---

### Official Review · Reviewer_R5Ru · 2023-10-31

**Soundness:** 2 fair
**Presentation:** 2 fair
**Contribution:** 2 fair
**Rating:** 3
**Confidence:** 4

**Summary:**

This paper presents Push, a Python library designed to ease the implementation of Bayesian deep learning (BDL) algorithms. A key feature of many BDL algorithms (e.g., Stein Variational Gradient Descent) is that during training, they evolve a *collection* of possible parameter settings for a neural network. Implementing this pattern efficiently can often require exploiting multiple GPUs to evolve particles concurrently. Push's key contribution is a new abstraction for concurrent programming across GPUs, based a centralized event loop that routes messages between concurrently executing particles. The paper provides a few demo implementations of BDL algorithms using the abstraction. In experiments, it demonstrates that Push's abstractions have only modest overhead (and can sometimes improve performance relative to naive baselines).

**Strengths:**

Overall, the authors appear to have created a useful library for an important task -- concurrent programming across many GPUs. I see several strengths of the work:

* This paper identifies a class of probabilistic models and inference algorithms that are underserved by current probabilistic programming languages. Both the algorithms (like SVGD) and the models (NNs with priors over extremely high-dimensional parameter spaces) are difficult to express in existing PPLs, and no PPLs (to my knowledge) provide explicit support for distributing inference computations across multiple devices.

* The Push library appears to address a real need: better programming models for concurrent programming across many GPUs.

* The paper provides examples of several BDL algorithms implemented using the Push concurrent programming model, and experiments demonstrating that they can make effective use of additional available devices.

**Weaknesses:**

**Relevance / value of the mathematical development.** Although it is described as a PPL for Bayesian deep learning, the actual technical content of Push does not appear to be specific to Bayesian deep learning (or any kind of probabilistic modeling and inference): it is a library for implementing algorithms that need to run various tasks concurrently across several GPUs, with some communication across tasks. The authors justify their BDL framing with a mathematical development (Sections 3.1, 3.3, 3.4; Appendix A) that treats the concurrent tasks as "particles" in a discrete approximation of a distribution. But the presentation (and why it matters) is somewhat unclear:
- Despite the insistence at several points that Push programs define *models* (e.g. in Sec. 3.3, or the top of Sec. 4), the code of a Push program appears to define neither a prior over the network weights nor a likelihood over the data. Rather, it directly implements an  algorithm, which may or may not be interpretable as a Bayesian inference algorithm in some model. The absence of a model makes it difficult to understand what the paper is saying at points. For example, the authors write, "The properties of the approximation (e.g., smoothness) depends on the interaction between particles encapsulated in a PD." What does smoothness mean here? What is being approximated? Or, in another spot, the authors write, "In general, a PD P(nn Θ) does not have a density... Assumptions that introduce densities open the possibility to apply more inference algorithms." But the term "PD" and the notation P(nn Θ) have previously been defined to refer to a discrete distribution of $n$ equally weighted particles, which can never have a (continuous) density function.
- Most practitioners of BDL will understand that inference is over the unknown parameters of a Bayesian network; the development of the "pushforward" view (inference is *really* over the random function implemented by the network) seems unnecessary. What is the value of this math for better understanding Push, or Push's design, or how to use the library effectively?

**Clarity of and motivation for the proposed technique.** The key automation that Push provides—managing the concurrent execution of multiple "particles" across GPUs—is not described with sufficient clarity. For example, the text says that "Each NEL contains a particle controller." But Figure 3 appears to show a single NEL with many particle controllers. As another example, there is no discussion of the order in which messages are processed or how the NEL decides which messages to process next. This seems consequential, because if the NEL is processing a message for which the receiving GPU is not available, it appears to "wait until the device is free" (instead of moving on to process more messages?). It is not clear whether the user can control (or whether the system attempts to optimize) placement of workers on different GPUs. More generally, I would like to see better motivation for the approach. What key challenges is Push addressing, and how is it addressing them? What space of designs was considered for the implementation, and what is good about the strategy you chose?

**Questions:**

1. Is there an intuition you can provide for what problem (if any) the program illustrated in Figs 1 and 2 is solving? (E.g., why has a user decided to write `_gather`?)

2. It seems somewhat unnatural to me that in your implementation of SVGD, the "leader" process needs to itself be a particle, rather than just a process that coordinates the other particles. At several points in the code you have to special-case the handling of the leader's parameters vs. everyone else's parameters. Why does Push require each "worker" in the concurrent algorithm to itself be a particle that is storing one set of weights for a neural network? Might it be more useful to present Push as a general-purpose library for concurrent programming on GPUs, with an application to BDL?

---

> ### Author Response · Authors · 2023-11-16
>
> We thank you for your time and thoughtful feedback. We refer you to our general response for additional clarification and are happy to answer any more questions.
>
> > Is there an intuition you can provide for what problem (if any) the program illustrated in Figs 1 and 2 is solving? (E.g., why has a user decided to write _gather?)
>
> We thank the reviewer for this question. The program illustrated in Figures 1 and 2 would be needed to implement the all-to-all communication pattern between neural networks required by SVGD. We use SVGD as an example of the densest communication pattern between neural networks that could be exhibited by a BDL algorithm. Deep ensembles are an example of a lack of communication between neural networks in a BDL context. We hope to investigate BDL algorithms and their characteristics  with sparser and dynamic communication patterns, i.e., communication dependent on the state of the neural networks, in future work. To more easily and systematically explore this, we were motivated to build a library that could easily express different communication patterns and heterogeneous computations across an ensemble of neural networks.
>
> > It seems somewhat unnatural to me that in your implementation of SVGD, the "leader" process needs to itself be a particle, rather than just a process that coordinates the other particles. At several points in the code you have to special-case the handling of the leader's parameters vs. everyone else's parameters. Why does Push require each "worker" in the concurrent algorithm to itself be a particle that is storing one set of weights for a neural network? Might it be more useful to present Push as a general-purpose library for concurrent programming on GPUs, with an application to BDL?
>
> We thank the reviewer for this question. It is possible to design push as a process + particles system so that we do not need to special case the "leader" process. The tradeoff in this case is that the process would need to introduce primitives itself for communicating with particles and particles would need special case handling communication from the process. We choose the design where everything is a particle so that communication can be handled uniformly. It might be possible to present Push as a general-purpose library for concurrent programming with applications to BDL, although the multi-threaded execution model operates at the level of particles/neural networks and not tensors. As we mention in the paper, it is possible that Push is useful for expressing other algorithms such as meta-heuristic algorithms, although we leave these investigations for future work.

---

> > ### Comment · Reviewer_R5Ru · 2023-11-22
> >
> > Thanks for clarifying these two questions! My assessment of the paper as it stands is unchanged, although I think it could possibly be developed into a valuable future submission about concurrent GPU programming, with BDL as a key use case.

---

### Official Review · Reviewer_G8dD · 2023-11-01

**Soundness:** 3 good
**Presentation:** 3 good
**Contribution:** 2 fair
**Rating:** 3
**Confidence:** 4

**Summary:**

This paper describes a Python library, "push", which can be used to orchestrate "particles" on a multi-GPU, single-host system, for experimentation in Bayesian deep learning. The paper provides timing results demonstrating that collective primitives such as all-reduce can provide speed benefits to SVGD relative to a baseline which does not replicate all-to-all. Extending to a multi-host distributed regime is left to future work.

**Strengths:**

*Originality*

The idea of a framework for orchestrating an ensemble of BNN samples is original. The particular realization here, with a Node Execution Loop and future-based async activity is perhaps novel (but I have questions)--although looked at through another lens, Java has had ThreadPoolExecutors and Futures, which facilitate async execution for quite a long time. Still, this might be new to the ML community, and the application to multi-accelerator, in Python, is much more relevant here than Java.

*Quality*

The experiments demonstrate reasonably well that push provides a low overhead approach to orchestrating various approaches to optimizing ensembles of NNs. User code is fairly readable (see appendix), though the async nature can make it a bit challenging to reason about what happens when.

*Clarity*

Exposition is clear, the work is well presented, the experiments are well explained. The basics of how the system is implemented are clearly explained, in particular noting "actor based" and "async-await" concurrency.


*Significance*

Having such a library available could be of interest to those in the Bayesian deep learning / BNN community. Arguably, a library that orchestrates asynchronous collective work across multiple accelerators is of interest to the community at large.

**Weaknesses:**

*Originality*

See questions below, e.g. why not straight torch, why not JAX?

I'm not sure that treating a single realization of a torch.Module weights sample as a "particle" and calling that module functor a pushforward is particularly novel. NNs already claim to be "function approximators" -- the very fact that torch.Module implements `__call__` suggests this was self-evident to the implementors.

*Quality*

The work is more a presentation of a new async parallel collectives orchestration system embedded in Python, built around torch.Module, than a presentation of novel results achieved using the system. In the vein of "experiment quality", it would be more compelling to me to also see some experiments along the lines of "here's something we could do now, that we couldn't have done before". Could we get BNN samples of ViT which are competitive with or superior to those in uncertainty_baselines?

*Clarity*

I thought the work was presented clearly, don't have substantial concerns here.

*Significance*

I have some hesitation about venue. The work feels a bit more like something I'd read from MLSys or JStatSoft than from ICLR. Put another way, I am unsure how many ICLR attendees would be impacted by learning about such a system. On the one hand, I want this system to be of much more general interest (along the lines of "pytorch async orchestrator"); on the other hand, I believe systems already exist that provide such solutions, so perhaps specialization to the space of BDL/BNNs is helpful. But then, reading the user code required to implement such BDL solutions atop Push, it's not clear how specialized to Bayes the system really is.

**Questions:**

The most immediate question to my mind is, why is straight torch insufficient to the task? With primitives such as model.to(torch.device("cuda:0")), model.to(torch.device("cuda:1")), etc., we can control the GPU residence of torch models, and data transfers, without an intervening library.

The next question that comes to mind is, why not jax.pmap? https://github.com/google/jax#spmd-programming-with-pmap
All of the BDL proposals tested in this work can be implemented as pure functions of weights, and JAX provides collectives such as psum and all-gather https://jax.readthedocs.io/en/latest/jax.lax.html#parallel-operators which seem to answer most of the useful parallel comms needs of push. Add to those a JIT compiler to manage memory and hide latency effectively.

Even if both of the above suggestions are for some reason deficient, it would be helpful to readers of the paper to elucidate reasons why.

---

> ### Author Response · Authors · 2023-11-16
>
> We thank you for your time and thoughtful feedback. We refer you to our general response for additional clarification and are happy to answer any more questions.
>
> > The most immediate question to my mind is, why is straight torch insufficient to the task? With primitives such as model.to(torch.device("cuda:0")), model.to(torch.device("cuda:1")), etc., we can control the GPU residence of torch models, and data transfers, without an intervening library.
>
> We thank the reviewer for this question. The advantage of using the library is primarily speed and convenience. The library automates the underlying threading mechanism that coordinates concurrent particle communication and execution on different GPUs for the user. This is why in Figure 4, the library implementation of Stein Variational Gradient Descent (which has all-to-all communication) is faster than the PyTorch only implementation even on 1 GPU across a range of particles. The library additionally creates views of parameters sets to help the user with memory management in cases like Stein Variational Gradient Descent where multiple neural network parameter sets are considered for the parameter update. The underlying multi-threaded implementation of the library also enables the library to utilize all GPU devices concurrently, whereas a straight PyTorch implementation does not. This can lead to speedups since the library implementation can access the GPU devices in any order and concurrently, whereas a PyTorch only implementation would access the GPU devices in the order specified in the original program and sequentially. While it is certainly possible to utilize multiple GPUs using PyTorch alone, one might need to write additional multi-threaded code or other modules to get similar performance.
>
> > The next question that comes to mind is, why not jax.pmap? https://github.com/google/jax#spmd-programming-with-pmap All of the BDL proposals tested in this work can be implemented as pure functions of weights, and JAX provides collectives such as psum and all-gather https://jax.readthedocs.io/en/latest/jax.lax.html#parallel-operators which seem to answer most of the useful parallel comms needs of push. Add to those a JIT compiler to manage memory and hide latency effectively.
>
> We thank the reviewer for this question. While this may change in the future, our understanding is that "Jax has limited support for Python concurrency" as stated in the documentation https://jax.readthedocs.io/en/latest/concurrency.html. Thus, it would be difficult to implement a concurrent library on top of Jax. It would be an interesting direction of future work to try to implement the library in Jax given its many useful features (such as JIT compilation) and nice engineering. Pmap and all-gather are useful constructs, but they operate at a lower level of abstraction compared to the level that particles are represented at. The concurrent execution semantics of particles in the library can provide the illusion of parallelism to scale across additional GPUs, but it also makes it easier to implement BDL algorithms where the communication patterns involved are not uniform and the computations that each particle perform may depend on the state of the particle. For example, an all-to-all communication pattern as in Stein Variational Gradient Descent is not scalable, and so we may want to explore BDL algorithms that have sparser and dynamic communication patterns instead that would not be easily expressed with pmap or all-gather.

---

> > ### Comment · Reviewer_G8dD · 2023-11-22
> > **Acknowledged**
> >
> > Thanks for the discussion.
> > I'm a bit surprised if PyTorch doesn't support async dispatch for multiple GPUs. I agree that would imply programming at a lower level, and thus Push can add value in terms of software design and implementation readability.
> >
> > I still think the package is positioned in an uncomfortable space between the more general purpose "async distributed orchestrator for PyTorch" and the more focused "parallel BDL toolkit": I think Push may already fulfill the general purpose tool in all but name/branding/positioning, and it seems to lack affordances that make parallel BDL very much easier or different from implementing such algorithms using a general toolkit.
> >
> > My point w.r.t JAX is that GPU primitives are by default dispatched asynchronously, such that the cross thread concurrency may no longer be needed: a single thread can schedule the work of all devices on a single host, because JAX Arrays serve as a Promise/Future. (But note, there may be some complexities w.r.t. cross-device copies, which would need resolving.)

---

> ### Author Response · Authors · 2023-11-22
>
> Thanks for your additional comments and feedback.
>
> > I'm a bit surprised if PyTorch doesn't support async dispatch for multiple GPUs.
>
> This is a subtle point that we were also surprised by. PyTorch does support async dispatch to GPUs in the sense that accessing GPU:0 and then accessing GPU:1 can continue executing on the CPU before GPU:0 and GPU:1 finish computing. However, if you do this from the same process, then PyTorch ensures that the request to access GPU:0 completes before the request to access GPU:1 which requires synchronizing the "multi-GPU" accesses. This is a performance cost which our system solves, since we remove the constraint that accessing GPU:0 must happen before GPU:1.
>
> > I still think the package is positioned in an uncomfortable space between the more general purpose "async distributed orchestrator for PyTorch" and the more focused "parallel BDL toolkit": I think Push may already fulfill the general purpose tool in all but name/branding/positioning, and it seems to lack affordances that make parallel BDL very much easier or different from implementing such algorithms using a general toolkit.
>
> We thank you for this feedback. We agree that there are more general use cases for the tool and mentioned some in the paper. We plan to explore these in future work, but have not tested it due to the constraints of a single paper.
>
> > My point w.r.t JAX is that GPU primitives are by default dispatched asynchronously, such that the cross thread concurrency may no longer be needed: a single thread can schedule the work of all devices on a single host, because JAX Arrays serve as a Promise/Future. (But note, there may be some complexities w.r.t. cross-device copies, which would need resolving.)
>
> Thank you for this explanation. This would be interesting to take a look into. However, as we explained above in PyTorch, if that single thread that JAX uses to schedule multi-GPU accesses guarantees that the Promise/Future obtained from GPU:0 and GPU:1 on a single thread must be produced in the order they are accessed, then there again would need to be synchronization across multi-GPU accesses.

---

### Official Review · Reviewer_omDv · 2023-11-09

**Soundness:** 3 good
**Presentation:** 4 excellent
**Contribution:** 4 excellent
**Rating:** 8
**Confidence:** 4

**Summary:**

This paper describes a new probabilistic programming library called Push, for composing together
Bayesian neural networks. Through an actor-inspired concurrency model where neural networks
accept and send a collection of particles. This allows the system to readily encode scatter-gather
patterns that are very amendable to working on GPUs as well as scaling linearly with multiple
GPUs.

**Strengths:**

This is a very unique and original approach for representing Bayesian neural networks. The design
is well-thought out and the experiments are equally thoughtful. The paper is clearly written and
I can easily see others using and extending this work.

**Weaknesses:**

The paper says this architecture can support many BDL algorithms but only SWAG and SVGD are presented.
While I think it's too much to ask experiments to be done on more algorithms, it would be nice if
at a high-level it can be shown how many of the most popular BDL algorithms would be supported by
the Push library.

**Questions:**

What BDL algorithms could be represented in this library?
What BDL algorithms would be challenging to use with this library?

---

> ### Author Response · Authors · 2023-11-16
>
> We thank you for your time and thoughtful feedback. We refer you to our general response for additional clarification and are happy to answer any more questions.
>
> > What BDL algorithms could be represented in this library? What BDL algorithms would be challenging to use with this library?
>
> We thank the reviewer for this question. It is possible to implement Monte-Carlo drop-out in this library in the standard way. It is possible to implement a Markov-Chain Monte Carlo algorithm in this library by using a particle to represent a proposal and keeping the particle if the proposal is accepted or reusing the particle and performing new proposals if the previous proposal is rejected. These proposals can use gradients. It is also possible to implement a variety of variational approximations such as Stein Variational Gradient Descent. It is also possible to implement message-passing style algorithms in the library by defining the appropriate send functions to update other particles parameters with knowledge of the calling particle's parameters/state. We hope to explore all of these avenues in future work. Algorithms such as backprop-by-bayes are not applicable in our representation since we do not directly define a distribution on parameters. We also hope that the representation of a communicating ensemble of particles may inspire the study of new BDL algorithms that have more complex communication patterns and use more particles.

---

> > ### Comment · Reviewer_omDv · 2023-11-22
> >
> > Thanks! Would it be possible if some of this answer made it into the paper?

---

> > > ### Author Response · Authors · 2023-11-23
> > >
> > > Yes, we have updated the paper to reflect some of this discussion.

---

### Author Response · Authors · 2023-11-16

We thank the reviewers for their time and thoughtful feedback. We address a common theme in the responses, namely: the connection between the library and BDL. Our motivation is to explore the representation of a Bayesian neural network (BNN) as an ensemble of neural networks/particles that may communicate the region of the posterior that it is exploring to coordinate with other particles. Thus, communication operationalizes conditional independence relationships in a BNN. This contrasts with an approach to defining a BNN by defining a prior on the weights of a NN and directly using the induced conditional independence structure. To explore the representation, we implement a library that can efficiently support this representation, identify the relationship between this representation and a BNN, validate that several existing BDL algorithms can be encoded in this representation, and experiment with its particle scaling properties (e.g., while holding effective parameter size constant to control for the cost of computation). We hope to use the library in future work to explore the benefits/limits of this representation, the key one being more complex and non-homogeneous communication patterns that operationalize conditional-independence relations in the BNN. We thank the reviewers for this question and will more clearly make this point to improve the context and motivation of the draft.

---

### Meta-Review · Area_Chair_1y7W · 2023-12-14

**Metareview:**

This paper primarily introduces two things: (1) a library which enables concurrency and communication in a multi-GPU setting in pytorch; (2) a framing of some popular Bayesian neural network inference algorithms into this setting, which demonstrates that the concurrent implementation benefits from improved performance.

There was some disagreement among the reviewers, with three firmly arguing to reject, and one firmly to accept.

However, even in the negative reviews, there was much praise for the methodology itself, its implementation, and the clarity of the writing. The primary issue is one of scope and venue fit. At the moment, the paper is much more focused on presenting the concurrency library, than on contributions related to BNNs or novelty in the learning algorithm itself.

As such, in its current form the paper probably isn't the best fit for ICLR.

Reviewers R5Ru and G8dD have a number of helpful suggestions for how to adapt the paper for a more general machine learning audience, or of alternative sort of venues.

**Justification For Why Not Higher Score:**

Probably not a good venue fit for ICLR (three of four reviewers rejecting)

**Justification For Why Not Lower Score:**

N/A

---

### Decision · Program_Chairs · 2024-01-16

Reject